# Can Frozen Transformers in Large Language Models Help with Medical Image Segmentation?

**Juntao Jiang**[1]                                         JUNTAOJIANG@ZJU.EDU.CN

[1] *Zhejiang University, 38 Zheda Road, Hangzhou, China*

**Yong Liu**[*1]                                          YONGLIU@IIPC.ZJU.EDU.CN

**Editors:** Accepted for publication at MIDL 2024

## Abstract

Transformer models shine in medical image segmentation by harnessing their self-attention mechanism to capture global information, thus boosting segmentation accuracy. Recent research has unveiled that large language models (LLMs), trained solely on text, surprisingly excel at visual tasks even without language, through a simple strategy: integrating a frozen transformer block from pre-trained LLMs as a direct visual token processor. This paper applies this approach to medical image segmentation by combining frozen transformer blocks with TransUNet. Experiments are conducted on BTCV, ACDC, ISIC 2017, CVC-ClinicDB, CVC-ColonDB and BUSI datasets, demonstrating some improvements compared to the baseline. The code will be released at: https://github.com/juntaoJianggavin/LLM4MedSeg.

**Keywords:** Medical image segmentation, Large Language Models, Frozen transformers, TransUNet

## 1. Introduction

Medical image segmentation refers to the process of partitioning medical images into meaningful regions or structures. By automatically delineating organs, tissues, tumors, and other anatomical structures with designed algorithms, computer-assisted segmentation facilitates quantitative analysis, patient-specific modeling, and personalized healthcare delivery. UNet(Ronneberger et al., 2015) and its variants hold a prominent position in deep learning-based medical image segmentation due to their combination of encoder-decoder structure and skip connections, effectively capturing multiscale features. Transformer-based UNet variants such as TransUNet (Chen et al., 2021), Swin-UNet (Cao et al., 2022) and Ds-transunet (Lin et al., 2022) exhibit remarkable capabilities in medical image segmentation owing to their overall architectures with self-attention mechanism, which adeptly captures global information and have stronger visual encoding capabilities.

Recent research (Pang et al., 2023) unveils the surprising effectiveness of large language models (LLMs), trained solely on text, as potent encoders for pure visual tasks by leveraging a frozen transformer block from pre-trained LLMs and another research (Lai et al., 2024) show that the LLM transformer block with global residual connections can boost medical image classification. This paper applies the method in (Pang et al., 2023) for medical image segmentation tasks, adding a frozen transformer block from LLaMA-7B to the encoder of TransUNet. Experiments on BTCV (Landman et al., 2015), ACDC (Bernard et al., 2018), ISIC 2017 (Codella et al., 2018), CVC-ClinicDB (Bernal et al., 2015), CVC-ColonDB (Bernal et al., 2012) and BUSI (Al-Dhabyani et al., 2020) datasets show that this method can lead to some improvements.

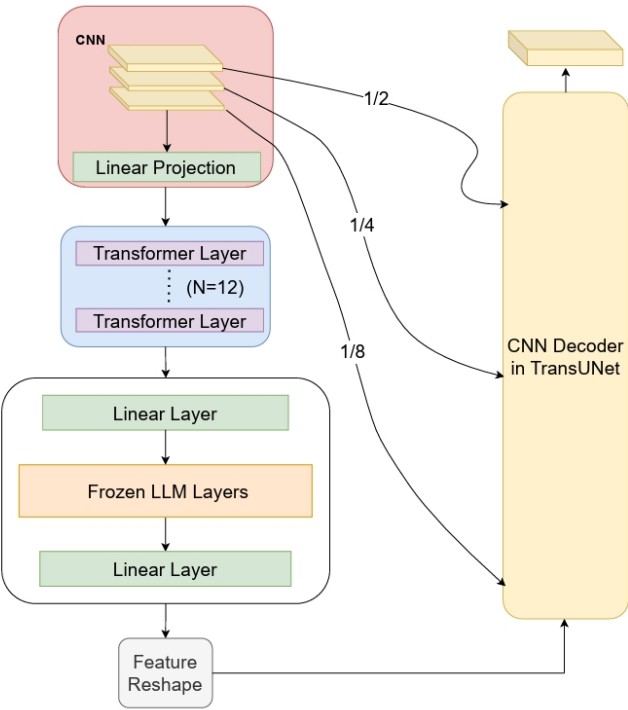

Figure 1: The architecture of TransUNet with frozen Transformer blocks in LLMs (LLM-TransUNet)

## 2. Method

The architecture of TransUNet with frozen Transformer blocks in LLMs (LLM-TransUNet) can be seen in Figure 1. A frozen transformer block from pre-trained LLaMA-7B (Touvron et al., 2023) is inserted after Transformer layers in TransUNet. As the feature dimensions are different between the transformers in TransUNet and the language model, two linear layers are used to align the dimensionality like the design in (Pang et al., 2023).

## 3. Experiments

### 3.1. Implementation Details

All experiments are conducted on Tesla PG500-216 GPU platform with 32G RAM. The resolution of input images is 256×256. For ISIC 2017 (Codella et al., 2018), CVC-ClinicDB (Bernal et al., 2015), CVC-ColonDB (Bernal et al., 2012) and BUSI (Al-Dhabyani et al., 2020), the total training epochs are 200, and the batch size in training, validation and testing is 8. The optimizer used is ADAM (Kingma and Ba, 2017). The initial learning rate is 0.0001 while the minimum learning rate is 0.00001. A CosineAnnealingLR (Loshchilov and Hutter, 2016) scheduler is utilized. Rotation by 90 degrees clockwise for random times, random flipping and normalization methods are applied for data processing and augmentation. The evaluation metrics in validation are *IOU* of the lesions and in testing are *IOU* and *Dice*

of the lesions. The loss function used is a mixed loss combining binary cross entropy loss and dice loss (Milletari et al., 2016):$\mathcal{L} = 0.5BCE(\hat{y}, y) + Dice(\hat{y}, y)$. For CVC-ClinicDB, CVC-ColonDB and BUSI, the testing sets are split from datasets with a ratio of 0.2. Then the validation sets are split from the rest of the datasets with a ratio of 0.2. The random states for splitting are all 41. ISIC 2017 has its own validation and testing sets.

For BTCV (Landman et al., 2015) and ACDC (Bernard et al., 2018), all settings except for the device specification and the input size of images, including the dataset preprocessing methods and training details, are all aligned with the configuration detailed in (Chen et al., 2021).

## 3.2. Results

The experimental results are shown in Table 1 and Table 2, which show that a frozen transformer layer in LLM can lead to improvement in medical image segmentation tasks. The explanation may be as the hypothesis in (Pang et al., 2023) mentioned – the incorporated LLM blocks may distinguish the informative tokens and amplify their effect. The mDice and the mHD95 in Table 2 refer to mean Dice and mean Hausdorff Distance 95, respectively.

Table 1: Comparison Experimental Results on ISIC 2017, BUSI, CVC-ClinicDB and CVC-ColonDB datasets

| Methods | ISIC 2017 | | BUSI | | CVC-ClinicDB | | CVC-ColonDB | |
|---|---|---|---|---|---|---|---|---|
| | IoU | Dice | IoU | Dice | IoU | Dice | IoU | Dice |
| TransUNet | 0.7355 | 0.8442 | 0.6323 | 0.7634 | 0.8500 | 0.9183 | 0.8526 | 0.9182 |
| LLM-TransUNet | **0.7532** | **0.8560** | **0.6436** | **0.7742** | **0.8525** | **0.9184** | **0.8728** | **0.9311** |

Table 2: Comparison Experimental Results on BTCV and ACDC datasets

| Methods | BTCV | | ACDC | |
|---|---|---|---|---|
| | mDice | mHD95 | mDice | mHD95 |
| TransUNet | 0.7845 | 30.4188 | 0.8988 | 2.6160 |
| LLM-TransUNet | **0.7869** | **25.7007** | **0.9029** | **1.2210** |

## 4. Conclusion

This paper explores integrating frozen transformer blocks from large language models (LLMs) into the TransUNet architecture for medical image segmentation. Experimental results demonstrate some improvements compared to the baseline, further proving that the LLM layers can be useful in visual recognition tasks. The designed module has the potential to become a plug-and-play component in medical image segmentation tasks. However, some datasets are quite small, thus the way of splitting datasets has a significant impact on final results. More experiments are needed to demonstrate the blocks can truly lead to improvements. Also, more baselines like Swin-UNet and Ds-transunet can be used for experiments to further explore this topic.

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
