# OpenReview forum: "Can Frozen Transformers in Large Language Models Help with Medical Image Segmentation?"
_MIDL.io/2024/Short_Papers — MIDL 2024 Short Papers_

### Official Review · Reviewer_p1jH · 2024-04-17

**Confidence:** 5
**Final Rating:** 4

**Review:**

The work investigates the effect of integrating frozen transformer blocks with Trans-UNet in the concept of medical image segmentation. The results are interestign and should be presented at the MIDL conference.
One weakness is that there is only one baseline architecture for comparison. Datasets need more explanation. Authors should address these during the conference.

---

### Decision · Program_Chairs · 2024-04-26

Accept